# miR-409-3p Regulates IFNG and p16 Signaling in the Human Blood of Aging-Related Hearing Loss

**DOI:** 10.3390/cells13181595

**Published:** 2024-09-23

**Authors:** Junseo Jung, Jeongmin Lee, Hyunsook Kang, Kyeongjin Park, Young Sun Kim, Jungho Ha, Seongjun So, Siung Sung, Jeong Hyeon Yun, Jeong Hun Jang, Seong Jun Choi, Yun-Hoon Choung

**Affiliations:** 1Department of Otolaryngology-Head and Neck Surgery, Cheonan Hospital, College of Medicine, Soonchunhyang University, Cheonan 31151, Republic of Korea; ing2094@naver.com (J.J.); minn5454@naver.com (J.L.); ssook4311@hanmail.net (H.K.); rudwls1289@naver.com (K.P.); 2Department of Biomedical Science, Soonchunhyang University, Asan 31538, Republic of Korea; 3Department of Otolaryngology, School of Medicine, Ajou University, Suwon 16499, Republic of Korea; jmyea@hanmail.net (Y.S.K.); jhflyflyfly@gmail.com (J.H.); thtjdwns123@gmail.com (S.S.); jhj@ajou.ac.kr (J.H.J.); 4Department of Medical Sciences, Graduate School of Medicine, Ajou University, Suwon 16499, Republic of Korea; ylem2s1u@gamil.com (S.S.); medicake@ajou.ac.kr (J.H.Y.)

**Keywords:** hearing loss, senescence, miRNA, IFNG, P16

## Abstract

Presbycusis, also referred to as age-related hearing loss (ARHL), is a multifaceted condition caused by the natural aging process affecting the auditory system. Genome-wide association studies (GWAS) in human populations can identify potential genes linked to ARHL. Despite this, our knowledge of the biochemical and molecular mechanisms behind the condition remains incomplete. This study aims to evaluate a potential protective tool for ARHL treatment by comparing human blood-based target gene-miRNA associations regulated in ARHL. To identify promising target genes for ARHL, we utilized an mRNA assay. To determine the role of miRNA in ARHL, we investigated the expression profile of miRNA in whole blood in ARHL patients with real-time polymerase chain reaction (RT-qPCR). A reporter gene assay was performed to confirm the regulation of candidate genes by microRNA. Through RT-qPCR validation analysis, we finally confirmed the relationship between ARHL and the role of the interferon-gamma (IFNG) gene. This gene can be regarded as an age-related gene. Through gene ontology (GO) analysis, it has been found that these genes are enriched in pathways related to apoptosis. Among them, IFNG induces an inflammatory response, apoptotic cell death, and cellular senescence. We found that miR-409-3p downregulates the expression of the IFNG in vitro. In addition, the downregulation of the IFNG by miRNA 409-3p promoted cell apoptosis and suppressed proliferation. In conclusion, our study produced gene signatures and associated microRNA regulation that could be a protective key for ARHL patients. IFNG genes and miR-409-3p should be investigated for their usefulness as a new biomarker for treatment modality.

## 1. Introduction

Inflammation has become a focal point in biomedical research on age-related conditions. As tissues age, the body undergoes a process called ‘chronic inflammation’. This low-grade, age-related inflammation, also known as ‘inflammaging’, progressively worsens over time [1]. Recently, the potential pathways of inflammation and its effect on age-related hearing loss (ARHL) have received more and more attention in the field of hearing research. Although the cochlea was traditionally thought to be an immune-privileged organ [2], recent research has demonstrated its susceptibility to systemic inflammation [3,4]. Prolonged activation of inflammatory cytokines, including IL-1α, IL-2, TNF-α, and NF-κB, which play critical roles in initiating, regulating, and amplifying immune responses, can penetrate cells within the inner ear, such as those in the endolymphatic sac [5,6]. Verschuur et al. conducted human studies that demonstrated a link between hearing thresholds and key serum biomarkers of low-grade inflammation [7]. They found associations between the average hearing thresholds in elderly individuals and their levels of IL-6, C-reactive protein (CRP), white blood cell counts, and neutrophil counts. Lowthian et al. also highlighted the potential therapeutic advantages of low-dose aspirin, a mild anti-inflammatory agent, for ARHL [8]. This finding adds to the growing body of evidence suggesting that chronic inflammatory processes are key mechanisms behind age-related conditions.

One promising method for identifying genetic variants that contribute to ARHL is the use of a genome-wide association study (GWAS), which screens a large number of genetic markers throughout the genome for associations with the trait of interest. Trpchevska et al. performed a meta-analysis of GWASs in a large sample of 723,266 individuals who had self-reported or clinically diagnosed hearing impairment [9]. The study identified a total of 48 loci that were significantly associated with hearing impairment, including 10 loci that had not been previously reported. Additionally, Ivarsdottir et al. demonstrated that a combined cohort from Iceland and the UK Biobank (totaling 121,934 individuals identified through pure-tone audiograms and self-reported hearing difficulties) revealed 21 new associations, including 13 rare variants [10]. This study performed analysis for RNA sequencing data from ARHL patients and found 11 genes related to ARHL.

MicroRNA(miRNA)s are a class of small, non-coding RNA molecules that occur naturally within organisms. Typically, miRNAs are 20–23 nucleotides long, and they function by binding to specific target messenger RNA (mRNA) transcripts via complementary base pairing [11]. The interaction between miRNAs and their target mRNA molecules can lead to gene silencing, translational repression, or target degradation. It is estimated that miRNAs have the potential to target up to 60% of all genes, and each miRNA can affect the expression of hundreds of target genes [12]. Recent studies have revealed differences in miRNA expression in the inner ear of two mouse strains, C57BL/6J and CBA/J, which are frequently used in research on aging and age-related diseases due to the unavailability of human-derived samples. In the context of age-related hearing loss (ARHL), 111 miRNAs in C57 mice and 71 miRNAs in CBA mice showed differential expression [13]. Among these, downregulated miRNAs were more common than upregulated ones. Zhang’s research further identified miRNAs associated with the aging of the cochlear duct’s lateral wall, in addition to the organ of Corti. Downregulated miRNAs, such as those from the miR-183 and miR-181 families, were linked to proliferation and differentiation pathways [14], whereas upregulated miRNAs, including members of the miR-29 and miR-34 families, were implicated in pathways that activate or enhance apoptosis, involving genes like p53, p27, and Bcl2 [15]. While miRNAs are recognized for their roles in regulating cellular proliferation, differentiation, and growth within the inner ear, their function in the blood concerning age-related hearing loss (ARHL) remains unestablished. Zhuoran et al. suggested that miR-155 selectively interacts with the 3′untranslated region (3′UTR) of Stat1, leading to the suppression of mRNA expression. Additionally, the absence of miR-155 leads to the depression of IFN-γ-related transcription factors, offering a potential explanation for the heightened IFN response observed in microglia lacking miR-155 [16]. Amado T et al.’s research uncovered that within CD8+ T cells, miR-181a constrains the production of IFN-γ by repressing the expression of the transcription factor Id2, consequently fostering the activation of the IFNG expression program [17]. In another study, the miRNA expression pattern was analyzed in melanoma cells undergoing IFN-γ-induced ferroptosis. Subsequently, Weinan et al. demonstrated that the upregulation of miR-21-3p enhances IFN-γ-triggered ferroptosis by promoting lipid peroxidation [18]. 

In the present study, we identified 11 potential genes by analyzing RNA sequencing from ARHL patients. Among the identified genes, we focused on the profiles of the IFNG gene, which is possibly involved in cochlear inflammation or ARHL. Moreover, we assessed the preventive role of miR-409-3p on the expression of the IFNG gene. 

## 2. Materials and Methods

### 2.1. Ethics Statement and Study Population 

The study protocol was approved by the Cheonan Hospital, Soonchunhyang University College of Medicine (IRB No. 2018-10-037). All participants gave written informed consent to take part in the study. The research adhered to the Declaration of Helsinki and received approval from the Institutional Review Board at Cheonan Hospital, Soonchunhyang University. Written consent was obtained from each participant. All the methods applied in the study were carried out in accordance with the approved guidelines. 

All enrolled participants underwent a physical examination of both ears and a hearing test by the pure tone audiometry in a quiet chamber by an interacoustic AC-40 (Interacoustics, Middlefart, Denmark) clinical audiometer according to the manufacturer’s instructions. The inclusion criteria for the normal hearing group (NH) were defined by the average hearing threshold ≤ 25 dB at 250 Hz, 500 Hz, 1000 Hz, 2000 Hz, and 40,000 Hz, and those of the ARHL group were defined by the average hearing threshold ≥ 40 dB at 250 Hz, 500 Hz, 1000 Hz, 2000 Hz, and 40,000 Hz, respectively. On the other hand, the exclusion criteria of the present study were a history of acute or chronic infection or symptoms of infection, hypertension, diabetes mellitus, chronic renal disease, or other otological diseases or surgery. 

### 2.2. Isolation of Peripheral Blood Mononuclear Cells (PBMCs) from Human Blood and the mRNA Differentiation Expression Assay 

Blood samples were collected from 12 individuals in the normal hearing group (28 ± 12.89 years, male:female = 5:7) and 12 individuals in the ARHL group (68 ± 8.56 years, male:female = 7:5). RNA extraction was performed using the QIAamp RNA Blood Mini Kit (Qiagen, Hilden, Germany), following the manufacturer’s protocol. The extracted RNA was quantified using a Nanodrop spectrophotometer, and these samples were subsequently used for RT-qPCR analysis. Collected blood was gently mixed in EDTA tubes and layered onto Ficoll-Paque (1:1 ratio). After centrifugation at 277 rcf for 30 min at room temperature without a break, the PBMC layer was collected, washed with 1× PBS (pH 7.4), and centrifuged at 277 rcf for 10 min again. The supernatant was discarded, and the cells were resuspended in 1× PBS for RNA extraction. RNA samples were analyzed using the NanoString nCounter Analysis System (NanoString Technologies, Inc., Seattle, WA, USA) [19], following the manufacturer’s guidelines. A 5 µL aliquot (containing 100–300 ng of RNA) was combined with 8 µL of Master Mix, which included the reporter CodeSet and hybridization buffer. Next, 2 µL of capture probe set were added. The mixture was thoroughly mixed and centrifuged. It was then incubated in a thermocycler (Bio-Rad Laboratories Inc., Hercules, CA, USA) at 65 °C for 16 h (with a maximum hybridization time of 48 h). After incubation, samples were transferred to a preparation station (NanoString Technologies, Inc.), where they were bound to a cartridge using the nCounter Master Kit. The preparation station, which accommodates 12 lanes, operated for approximately 2.5 to 3 h. Following this, the cartridges were moved to a digital analyzer (NanoString Technologies, Inc.) for analysis. The digital analyzer then scanned the cartridges across 555 fields of view.

### 2.3. Quantifying Gene Expression and Differentially Expressed Gene Analysis

The initial step in quantifying gene expression values involved the analysis of normalization using the geNorm algorithm [20] within nCounter Advanced Analysis ver2.0.115 (NanoString Technologies, Inc.) [21]. Subsequently, differentially expressed genes between the chosen biological conditions were assessed using default parameters. To analyze expression profiles across samples, unsupervised clustering of normalized expression values for a chosen subset of differentially expressed genes was conducted using nCounter Advanced Analysis version 2.0.115. This software was also employed to create plots showing gene expression values and a volcano plot for fold-changes in expression. To understand the biological roles of differentially expressed genes across various conditions, a gene set enrichment analysis was performed. This analysis involved comparing the differentially expressed genes with functionally categorized gene sets, including those related to biological processes in gene ontology (GO), Kyoto Encyclopedia of Genes and Genomes (KEGG) pathways, and other functional gene sets. g:Profiler [22] and clusterProfiler [23] were employed for this purpose.

### 2.4. Validation of Target Gene Expression (RT-qPCR)

To validate the expression of candidate genes identified from mRNA sequencing results, RT-qPCR was employed with whole blood of the NH and ARHL patients (Table 1). The reverse transcription reactions were carried out according to the manufacturer’s protocol using the miScript II RT Kit (Qiagen, Hilden, Germany). To perform RT-qPCR, 2 µL of 5× miscript HiFlex Buffer, 1 µL of 10× miScript Nucleics Mix, 1 µL of miScript Reverse Transcriptase Mix, and 1 µg of Template RNA were mixed. Then, RNase-free water was added to make a total volume of 10 µL. RT-qPCR was conducted using an ABI StepOnePlus instrument (Thermo Fisher Scientific Inc., Waltham, MA, USA) and associated software. RT-qPCR was performed with initial denaturation at 95 °C for 10 min, followed by 40 amplification cycles. Each cycle consisted of 30 s at 95 °C for denaturation, 30 s at 60 °C for annealing, and 1 min at 72 °C for extension. Glyceraldehyde-3-phosphate dehydrogenase (GAPDH) served as the internal control, and the analysis was performed using SYBR Green. The expression levels of each target gene relative to GAPDH were measured using the 2-ΔΔCt method. To identify target genes of miRNAs, experimentally validated miRNA-target gene pairs were retrieved from MirTarBase [24]. Additionally, mature miRNA sequences were obtained from the miRBase database [25].

### 2.5. Gene Ontology Enrichment Analysis for miRNA Related to IFNG 

We conducted KEGG pathway enrichment analysis and gene ontology enrichment analysis (covering Biological Processes, Cellular Components, and Molecular Functions) to determine the functions of differentially expressed genes (DEGs). This was accomplished using the EnrichR database “http://amp.pharm.mssm.edu/Enrichr/ (accessed on 17 November 2023)” [26]. The Database for Annotation, Visualization, and Integrated Discovery (DAVID) “https://david.ncifcrf.gov/ (accessed on 17 November 2023)” was utilized for differential gene analysis, pathway enrichment, and biological annotation. The most significantly enriched terms were identified based on a *p*-value threshold of <0.05, using Fisher’s exact test. Clustering analysis was visualized with a heatmap generated through the web tool Morpheus “https://software.broadinstitute.org/morpheus (accessed on 19 May 2024)”.

### 2.6. UTR Vector Construction and miRNA

IFNG Human 3′UTR clone (#SC208099, Origene Technologies, Rockville, MD, USA) and miRNA 409-3p (HmiR0244-MR04-10, GeneCopoeia, Rockville, MD, USA) were employed for transfection. The IFNG 3′UTR clone was utilized with the pMirTarget 3′UTR assay vector, which serves as a cloning vector for 3′UTR clones for miRNA target validation. The assay reporter is luciferase, and *E. coli* DH5α: selection can be achieved through kanamycin resistance. The insert size is 615 bp. miRNA-409-3p was constructed using the pEZX-MR04 vector, with puromycin as the selection marker, and its mature sequence is represented as gaauguugcucggugaaccccu.

### 2.7. Cell Culture and Transfection 

The House Ear Institute-Organ of Corti 1 (HEI-OC-1) cells of mice were incubated at 37 °C with a 5% concentration of carbon dioxide and harvested 10% FBS DMEM (Dulbecco’s Modified Eagle’s Medium, Welgene, Gyeongsan, Republic of Korea). The cells were maintained within a range of 70–90% confluency during the experiments. HEI-OC-1 cells were seeded into a 6-well plate at a density of 1.5 × 10^5^ cells per well. After seeding, 24 h later, transfections were performed using Lipofectamine 3000 (Invitrogen, Waltham, MA, USA) with the IFNG 3′UTR clone (Origene Technologies) and miRNA-409-3p (GeneCopoeia). The mixture was incubated at room temperature for 15 min according to the manufacturer’s protocol and then applied to the cells. Subsequently, the cells were incubated at 37 °C for 1–2 days.

### 2.8. Dual-Luciferase Reporter Assay

We used the Dual-Luciferase Reporter assay system (DLR assay system, Promega Corporation, Madison, WI, USA) to conduct dual-reporter assays with pmiRGLO-based reporter systems. The DLR assay system measured luciferase activity in cells co-transfected with 3′UTR-IFNG vectors and the pmiRGLO control vector. Twenty-four hours post-transfection, the growth medium was removed, and cells were gently rinsed with phosphate-buffered saline. Next, 200 µL of passive lysis buffer (Promega, USA) were added to each well, and the plates were gently rocked for 15 min at room temperature. The cell lysates were then collected for the Dual-Luciferase Reporter assay. In white opaque 96-well plates (Corning Inc., New York, NY, USA), 20 µL of cell lysates were transferred. The firefly and renilla luciferase activity assays were conducted sequentially on the cell lysates in each well. To normalize for cell numbers and transfection efficiency, the pmiRGLO vector was used as an internal control, and the relative luciferase activity was calculated as the ratio of firefly/renilla luciferase activity.

### 2.9. Western Blot Analysis

Cells were lysed using RIPA buffer, which consisted of 25 mM Tris-HCl (pH 7.6), 150 mM NaCl, 1% NP-40, 1% sodium deoxycholate, and 0.1% SDS. Proteins were quantified using a Pierce BCA Protein Assay Kit (Thermo Fisher Scientific Inc.). Cell lysates containing equal amounts of protein were loaded onto a 4–15% SDS polyacrylamide precast gel (Bio-Rad Laboratories Inc.) and subjected to electrophoresis at 70 V for 3–40 min, followed by 110 V for 5–60 min. The following protein replication was repeated three times. Additionally, the loaded proteins were transferred to the PVDF membrane and blocked with 5% BSA for 1 h at room temperature. After blocking, the membranes were incubated overnight at 4 °C, including gentle shaking with primary antibodies (1:1000, IFN-γ, p16 antibody, Cell Signaling Technology, Danvers, MA, USA). Additionally, the membranes were incubated at room temperature for 2 h with horseradish peroxidase (HRP) conjugated secondary antibodies (1:3000, Cell Signaling, USA). Each protein was normalized to β-actin (1:5000, Cell Signaling, USA), and finally, blots were detected using Super Signal West Atto Ultimate Sensitivity Chemiluminescent Substrate (Thermo Fisher).

### 2.10. Statistical Analysis

Statistical analysis was performed using GraphPad Prism 9.5.0, Graphad Software Inc., Boston, MA, USA. All statistical analyses were conducted using one-way ANOVA (and nonparametric). Graphs were presented as mean ± SD, and significance among groups was determined using the Bonferroni post hoc test. Values were considered significant when *p* < 0.05.

## 3. Results

### 3.1. Inverse Correlation of IFNG and Putative Target miRNAs in the Whole Blood between NH and ARHL Groups

We analyzed the expression profiles of our target genes using Nanostring-mRNA assay data obtained from PBMCs of individuals with ARHL. Figure 1A presents a heatmap depicting the differential expression in mRNA levels. Figure 1B is a Venn diagram showing the number of genes up or downregulated in patients with age-related hearing loss based on mRNA assays in human blood. Figure 1C portrays a volcano plot representing the expression levels of candidate genes, while Figure 1D presents a scatter plot.

To validate the mRNA assay results of selected DEGs, RT-qPCR was conducted using whole blood samples. Finally, among 11 genes, five genes showing correlation with RNA sequencing results were found. In particular, the expression levels of the IFNG gene exhibited statistically significant differences (adjusted *p* ≤ 0.05) between individuals with ARHL and their corresponding NH (Figure 2). As depicted in the figure, the expression levels of IFNG were found to be higher in whole blood obtained from patients with ARHL compared to NH (normal hearing group).

By specifically focusing on IFNG genes (Figure 3A), we analyzed their expression as fold-changes in mRNA levels between the NH (*n* = 12) and ARHL (*n* = 12) groups. RT-qPCR was employed to detect IFNG expression in whole blood (Figure 3B), revealing noteworthy upregulation in IFNG expression in the ARHL group compared to the NH group. For further investigation, we conducted RT-qPCR analysis to assess the expression of miR-409-3p in whole blood samples from individuals with ARHL. Figure 3C demonstrates the significant downregulation of miR-409-3p expression in the ARHL group compared to the NH group, with the relative levels normalized to U6. Statistical analysis revealed a * *p* < 0.05, indicating the significance of the observed differences. 

### 3.2. IFNG Is a Direct Target of miR-409-3p in HEI-OC-1 Cells

To identify the potential regulators of IFNG expression in ARHL, we conducted bioinformatics analyses. We used databases, such as miRTarBase and miRbase, to explore target miRNAs for IFNG. Our investigation revealed a validated binding target site for miR-409-3p in the 3′UTR of IFNG (Figure 4A). Next, we employed human IFNG vectors (pmirGLO vector) with the firefly luciferase gene fused downstream. RT-qPCR analysis indicated that miR-409-3p expression levels were significantly reduced in HEI-OC-1 cells from the ARHL group compared to the NH group (Figure 3C). Additionally, luciferase reporter assays showed a notable decrease in luciferase activity in HEI-OC-1 cells co-transfected with the miR-409-3p vector and the 3′UTR of IFNG reporters. Importantly, off-target miRNA did not negatively impact the putative binding site of miR-409-3p (Figure 4B).

### 3.3. miR-409-3p Regulates IFNG and p16 Expression

As is evident from the Western blot results (Figure 5), miRNA-409-3p regulates the expression of the IFNG gene. To provide more robust evidence in this study, HEI-OC-1 cells were transfected with 1000 ng of the IFNG gene as an inducer and co-transfected with varying concentrations of miRNA-409-3p. The results of IFNG gene detection showed that as the concentration of miRNA-409-3p increased, the expression of the IFNG gene decreased. Therefore, it can be concluded that the overexpression of miRNA-409-3p downregulates IFNG. Additionally, the detection of the senescence-related marker, p16, also demonstrated that as the concentration of miRNA-409-3p increased, the expression of the p16 protein decreased. The schematic diagram (Figure 5D) summarizing our study revealed a significant upregulation of IFNG in PBMCs associated with ARHL. Our findings demonstrated that miR-409-3p, possessing a binding site in the 3′UTR of IFNG, induced the downregulation of IFNG and p16 signaling. This finding indicates that miR-409-3p could serve as a potential biomarker for ARHL, which justifies further research and investigation.

## 4. Discussion

Through RNA sequencing and Western blot analysis, we confirmed that miR-409-3p reduces the expression of IFNG and p16, highlighting its significant role in ARHL. These findings suggest that different approaches are required for preventive treatments or methods to address the various mechanisms of age-related hearing loss. Additionally, analysis of single nucleotide polymorphism (SNP) and exploration of potential genetic factors related to ARHL could be valuable directions for future research in this area.

IFN-γ functions as a cytokine with critical involvement in maintaining tissue homeostasis, orchestrating immune and inflammatory responses, and contributing to the surveillance of tumor immunity [27]. There are several studies on IFNG and senescence. Prolonged in vitro exposure of cells to either IFN-β or IFN-γ induces premature senescence through the activation of reactive oxygen species (ROS) and the DNA damage response (DDR) signaling pathways [28]. Volpe, Eugene A. et al. provided evidence supporting the active involvement of IFN-γ in the age-related loss of conjunctival goblet cells, providing insights into the complex cytokine interactions underlying the development of dry eye [29]. In addition to senescence, there are studies related to hearing loss and cochlear. The frequency of T cells capable of producing IFN-γ was significantly increased in 25% of PBMCs from patients with autoimmune sensorineural hearing loss [30]. Moon, Sung K et al. illustrated that through JAK1/2-STAT1 signaling, IFN-γ heightens the susceptibility of HEI-OC-1 cells to TNF-α-induced cytotoxicity. This underscores the role of IFN-γ in sensitizing cochlear cells to the cytotoxic effects of TNF-α [31]. In the present study, a transcriptome analysis was conducted on human blood samples to investigate ARHL. These lists include six genes that exhibited at least two-fold upregulation and five genes that exhibited at least two-fold downregulation between the ARHL and NH groups, with adjusted *p*-values less than 0.05 in both populations. Through gene ontology analysis, it was found that these genes are enriched in pathways related to the immune response and apoptosis. Among them, we chose the IFNG gene because it induces an inflammatory response, apoptotic cell death, and cellular senescence.

miRNAs are small non-coding RNAs that regulate gene expression and have been implicated in various cellular processes. Several papers have been published on the topic of miRNA and ARHL. Verónica Miguel et al. reviewed the current literature on how environmental factors, such as noise exposure and stress, can alter miRNA expression patterns in the inner ear, leading to hearing loss. They also discussed the potential for miRNAs as biomarkers for hearing loss and as therapeutic targets for preventing or treating hearing loss [32]. Zhang et al. focused on identifying miRNAs involved in ARHL by analyzing the expression of miRNAs in the organ of Corti, the sensory epithelium of the inner ear. The authors used a mouse model of ARHL and found that the expression of several miRNAs is altered in the organ of Corti during the progression of hearing loss. They suggested that these miRNAs may play a role in the degeneration of the organ of Corti and offer potential targets for therapeutic interventions [13]. miR-34a and miR-155 contribute to ARHL by targeting the SIRT1 deacetylase. This 2013 study published in *Nature Communications* investigated the role of miRNAs in ARHL using a mouse model. The study identified miR-34a and miR-155 as key regulators of the SIRT1 deacetylase, which plays a role in the maintenance of inner ear function [33]. Lee et al. reported that hsa-miR-409-3p was upregulated in senescent endothelial progenitor cells (EPCs), where it functioned as a negative regulator of angiogenesis by targeting the PPP2CA gene and modulating the PP2A/p38 signaling pathway [34]. Our findings from human PBMCs support the potential of hsa-miR-409-3p as a biomarker for human aging. Another study demonstrated that the concurrent upregulation of miR-30a-5p and miR-409-3p, along with the downregulation of miR-30a-3p and miR-181a-5p, can replicate cellular senescence in vascular endothelial cells [35]. This approach offers a single-step method for inducing senescence, as opposed to the more time-consuming process of replicative passage senescence. The present study showed that miR-409-3p was confirmed to regulate the target gene, IFNG. The expression of the IFNG gene and miR-409-3p were found to be directly inversely correlated to each other. To date, IFNG has been known as a target gene of miR-409-3p. It is known that the decrease in DGCR8 in ITP leads to the downregulation of miR-409-3p, which is related to the upregulation of IFNG [36]. In the present study, we discovered for the first time that miR-409-3p is a target miRNA of IFNG in hearing loss. The function of miR-409-3p in ARHL may be more complex than expected, as each miRNA is known to have the ability to regulate multiple genes, and many other miRNAs are also involved in ARHL. Therefore, further research on the relationship seems necessary.

The occurrence of aging and many diseases can be attributed to chronic inflammation. Interferons play a crucial role in the immune system’s defense against viral infections. Interestingly, in mammalian hosts, various tissues and organs that undergo aging continue to show alterations resulting from the activation of the interferon pathway [37]. The regulation of senescence involves the following two primary pathways: p21 and p16 [38,39]. A previous study revealed that p16 is the main factor responsible for accelerated senescence in vitiligo melanocytes [40] or normal melanocytes at high passage levels [41]. p16 INK4a forms a direct binding interaction with CDK4/6, preventing the phosphorylation of the retinoblastoma tumor suppressor (Rb) and initiating growth arrest from the G1 to S phase in the cell cycle [42,43]. The function of p16 INK4a involves the inhibition of the activity of cyclin-dependent kinase (CDK), and prolonged expression of p16 INK4a results in the induction of cellular senescence. Consequently, precise control of p16 INK4a is imperative to uphold a meticulous equilibrium between the suppression of tumors and the promotion of senescence [44]. According to a study conducted by Almontashiri, Naif A M et al., in every cell type examined, with the exception of HUVECs, where interferon-γ did not alter expression, the upregulation of p16 and p15 was observed irrespective of the 9p21.3 genotype [45]. In another study, it was referred to as interferon-γ–inducible protein p16 [46]. In the present study, we found that the expression of p16 in HEI-OC-1 cells was affected by IFN-γ and miR-409-3p treatment. We hypothesized that the induction of senescence in HEI-OC-1 cells by IFN-γ was mediated through p16. However, it is not clear by which mechanism and pathway miR-409-3p regulates p16 signaling, and further research on animal models is needed. 

Hearing disorders can be caused by a range of ototoxic factors, including prolonged and loud noise, vibrations, gentamicin, ionizing radiation, cisplatin, high doses of aspirin, bacterial and viral infections, genetic mutations, and aging. As miRNAs are highly expressed in the inner ear, variations in the levels of miRNAs could potentially contribute to the pathogenesis of hearing disorders. However, signals that cause these changes in the expression of miRNAs remain unclear. Increased oxidative stress and inflammation are major contributors to the pathogenesis of hearing defects induced by ototoxic agents [47]. These conditions are also known to play a central role in neurodegenerative diseases, including Alzheimer’s disease. Reactive oxygen species (ROS) and pro-inflammatory cytokines are likely involved in mediating the harmful effects of these conditions on hair cells by modifying miRNA expression. This is supported by the fact that ROS and pro-inflammatory cytokines alter the expression of miRNAs in non-auditory neurons [48], leading to their damaging effects. Additionally, several studies have demonstrated that immune cells are present in the human inner ear and that activation of the systemic immune system can contribute to cochlear degeneration, potentially resulting in permanent hearing loss [49,50,51]. Inflammation in the cochlea may arise from macrophages within the inner ear, either recruited from circulating monocytes originating in bone marrow or possibly resident cells [52,53]. Consequently, this study concentrated on examining gene expression alterations in peripheral blood leukocytes of ARHL patients, suggesting that these systemic immune changes may be linked to ARHL.

The present study is constrained by a restricted number of clinical samples and relies on in vitro experimental data. Consequently, future research employing animal models is essential to validate and extend these findings. Additionally, potential age-related effects, which were not entirely controlled in our experiments, should be considered when interpreting the results. Nonetheless, this study holds substantial academic significance as it utilizes human whole blood samples and human-derived materials. Notably, it identifies a noteworthy inverse relationship between IFNG and miRNA 409-3p levels, highlighting their potential roles in the pathogenesis of hearing loss. Moreover, the study provides insights into the involvement of p16 in auditory dysfunction. 

## 5. Conclusions

We suggest that the potential role of miR-409-3p regulates IFNG expression by the p16 signaling pathway in ARHL patients. These findings suggest that different developmental approaches are necessary for preventive treatments or strategies to address age-related hearing loss.

## Figures and Tables

**Figure 1 cells-13-01595-f001:**
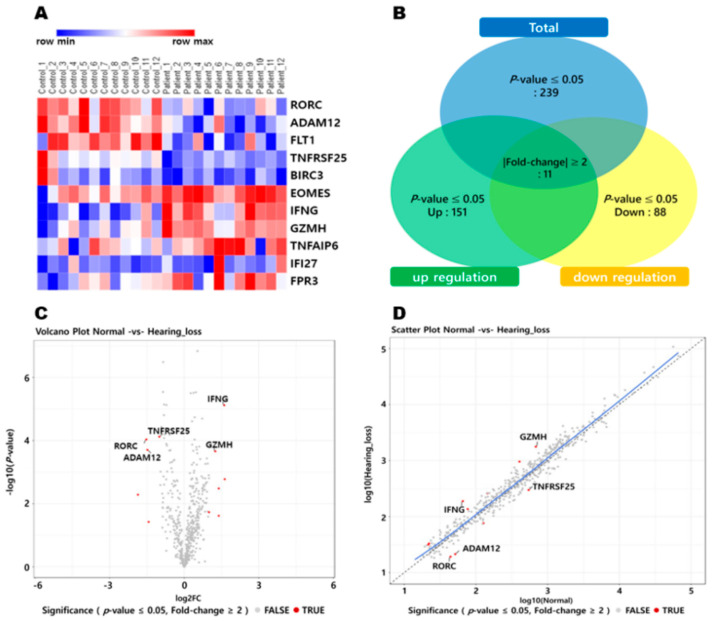
(**A**) The heatmap for the mRNA assay displays the 11 genes with the greatest increases or decreases in ARHL. The color gradient, ranging from blue to red, represents low to high levels of expression (hierarchical clustering is shown on the left of the heatmap). (**B**) Venn diagram illustrates the number of genes either up or downregulated in ARHL patients based on the mRNA assay in human blood. There are 11 genes included (*p*-value ≤ 0.05 and |Fold Change| ≥ 2). (**C**) A volcano plot was generated to visually represent the expression levels of candidate genes. The red dots represent a TRUE value for significance in the plot. (**D**) Scatter plot. Abbreviations used are as follows: RORC (Register of Registrable Controllers), ADAM12 (ADAM metallopeptidase domain 12), FLT1 (Fms-related receptor tyrosine kinase 1), TNFRSF25 (TNF receptor superfamily member 25), BIRC3 (Baculoviral IAP repeat-containing 3), EOMES (Eomesodermin), IFNG (interferon gamma), GZMH (Granzyme H), TNFAIP6 (TNF alpha-induced protein 6), IFI27 (interferon alpha inducible protein 27), and FPR3 (Formyl peptide receptor 3).

**Figure 2 cells-13-01595-f002:**
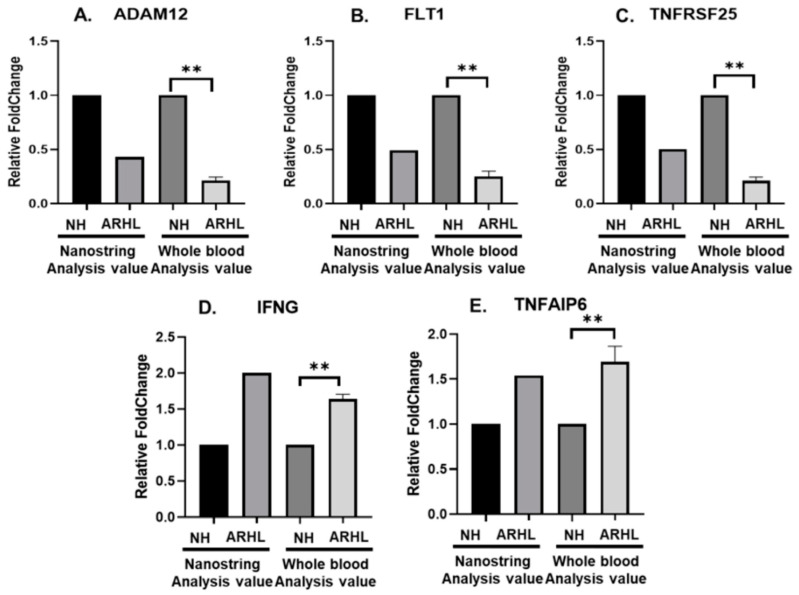
(**A**–**E**) Gene sets using real-time RT-qPCR. The samples included whole blood from the NH (normal hearing group) and ARHL. Gapdh served as an internal control, and the relative expression levels of ADAM12, FLT1, TNFRSF25, IFNG, and TNFAIP6 were normalized to Gapdh. Statistical significance was determined with ** *p* < 0.01. (**A**) Relative expression levels of ADAM12. (**B**) Relative expression levels of FLT1. (**C**) Relative expression levels of TNFRSF25. (**D**) Relative expression levels of IFNG. (**E**) Relative expression levels of TNFAIP6. Abbreviations used are as follows: NH (normal hearing group), ARHL (age-related hearing loss), ADAM12 (ADAM metallopeptidase domain 12), FLT1 (Fms-related receptor tyrosine kinase 1), TNFRSF25 (TNF receptor superfamily member 25), IFNG (interferon gamma), and TNFAIP6 (TNF alpha-induced protein 6).

**Figure 3 cells-13-01595-f003:**
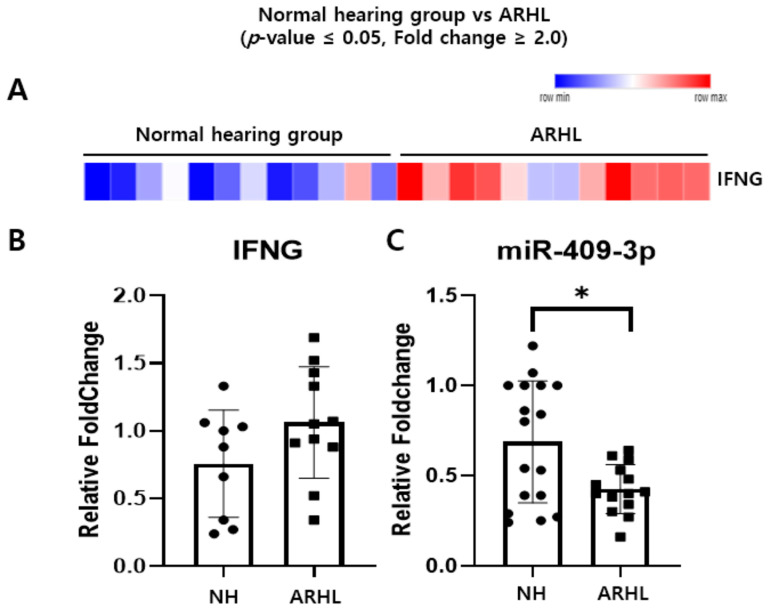
Inverse correlation between IFNG and potential target miRNAs in the whole blood of normal hearing (NH) and age-related hearing loss (ARHL) samples. (**A**) Heat map illustrating mRNA assay differential expression data. Heat map of IFNG genes involved expressed as the fold-change in mRNA levels in the NH and ARHL groups (NH: *n* = 12, ARHL: *n* = 12). (**B**) The expression of IFNG was further validated using real-time RT-qPCR in whole blood (NH: *n* = 9, ARHL: *n* = 11) (**C**) Additionally, the expression of miR-409-3p was measured in whole blood using real-time RT-qPCR, with U6 snRNA used as an internal control. The levels of miR-409-3p were normalized relative to U6 (NH: *n* = 17, ARHL: *n* = 14). Statistical significance is denoted by * *p* < 0.05. Abbreviations used are as follows: NH (normal hearing group), ARHL (age-related hearing loss), and IFNG (interferon gamma).

**Figure 4 cells-13-01595-f004:**
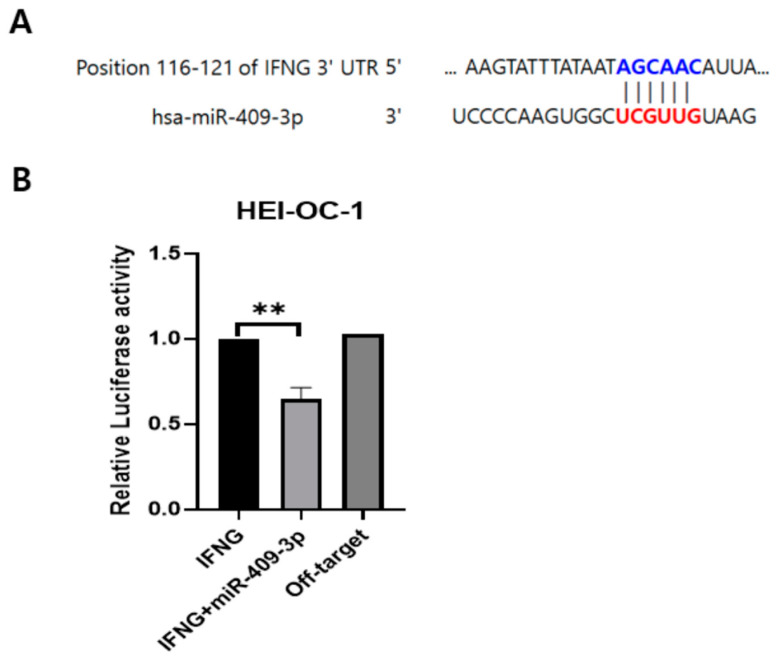
Results of the dual luciferase reporter assay of miRNA and its target gene. (**A**) Prediction of the 3′UTR sequences of the IFNG gene, highlighting their binding sites for miR-409-3p. (**B**) Results of the dual luciferase assay showing the interaction between miR-409-3p and the 3′UTR of IFNG in HEI-OC-1 cells. ** *p* < 0.01. Abbreviations used are as follows: IFNG (interferon gamma).

**Figure 5 cells-13-01595-f005:**
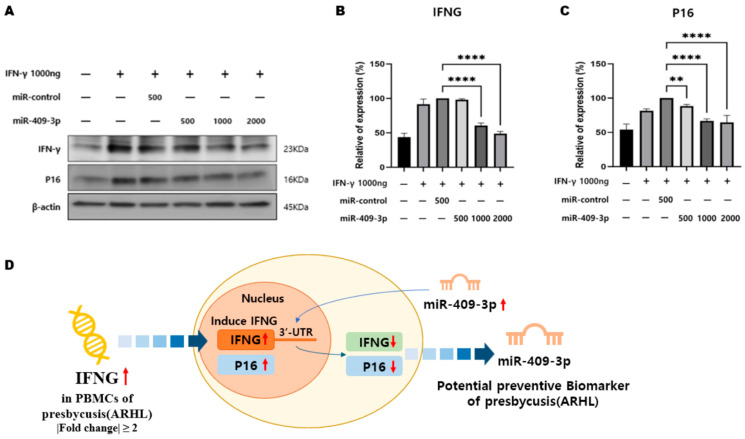
Western blot analysis was conducted as follows: (**A**) HEI-OC-1 cells were co-transfected with 1000 ng of IFNG, miR-control, and varying concentrations (ng) of miR-409-3p, as illustrated. The expression levels of IFNG, p16, and β-actin were subsequently detected using their respective antibodies. (**B**) Quantification of IFNG expression in each lane was performed using β-actin as a normalization control. (**C**) The p16 expression level in each lane was quantified using β-actin as a normalization control. Statistical analysis revealed a high level of significance with ** *p* < 0.01 and **** *p* < 0.0001. (**D**) Summary schematic diagram of IFNG, miR-409-3p, and P16 signaling in ARHL. Among the differential expression genes obtained through nanostring, IFNG was upregulated with a fold change value of 2 or more in PBMCs with age-related hearing loss. miR-409-3p, which has a binding site in the 3’UTR of IFNG, targeted and downregulated IFNG in HEI-OC1 cells overexpressing IFNG. Additionally, the expression of P16INK4a, a well-known senescence marker, was downregulated. Therefore, this suggests that miR-409-3p may be a potential biomarker for ARHL. Abbreviations used are as follows: IFNG (interferon gamma).

**Table 1 cells-13-01595-t001:** The sequence of primers used in the RT-qPCR experiments.

Name	Forward Primer	Reverse Primer
ADAM12	CAGGAAGGCTTGTTGTGCTT	TTGGGATCTCTTGAGCTGCA
FLT1	GACTGACAGCAAACCCAAGG	TAGATGGGTGGGGTGGAGTA
TNFRSF25	TTCTAGCACCTCCTGACAGC	ACAGGAGAATGGGGTCAAGG
IFNG	GGGGCTCAGTTTCCTCATCT	TAGAGACTTGCAGTGGGGTG
TNFAIP6	TACTGGGAAGTTTGGCGCTA	GTTCCTCTCCCTTCTCCCAC

## Data Availability

The original data included in the figures are shown in the article. For further inquiries, please contact the corresponding author.

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
