# Peer review of "miR-409-3p Regulates IFNG and p16 Signaling in the Human Blood of Aging-Related Hearing Loss"

_cells, 2024, doi:10.3390/cells13181595_

Round 1
Reviewer 1 Report
Comments and Suggestions for Authors
The authors have taken a small cohort of individuals with age related hearing loss and compared to much younger individuals with normal hearing levels. They have identified a changed in mRNA expression in 11 genes using the nano string and further validated a submit set of this using RTqPCR. They have then focused on one gene, IFNG to show in vitro using a reporter cell assay that miRN409 can reduce the protein expression of IFNG and P16.
The authors have provided sufficient information in their introduction but need to include more methodology for reproducibility. This could be aided by the inclusion of more cited references or a supplementary.
I note that the figures the authors have presented are of high resolution but also provide the strong basis of their scientific argument which is clearly articulated in their discussion. I found the scientific argument to be clear and it was supported by appropriate references. Publishing this data would be beneficial to the field. The authors have identified the pitfalls in their final discussion paragraph. I agree with their conclusions (Lines 396-403).
For a future study, I'm wondering if there are any SNPs in your cohort that segregate with the 11 genes in the ARHL individuals.
My suggested editing corrections:
Line 29 - Use correct acronym - RTqPCR. RT PCR refers to reverse transcription PCR.
Line 30 - write microRNA in full or add to introduction - line 69.
Line 32 - Remove a "s". e.g. Change "as age -related genes" to "age-related gene"
Line 35 - remove "in vitro study" and add "in vitro to the end of the sentence.
Line 67 - I would not call it a GWAS as the cohort is too low and you have used RNA and not a genetic marker.
Line 68 - add a reference or "this study"
Line 96 - remove space at the start of the sentence
Line 108 - use lower case "enrolled"
Line 113 - use "on the other hand" - i.e. remove the "s"
Line 116 - alternative wording in the title? ie. mRNA differentiation expression assay, rather than just using mRNA assay.
Lines 121-127 - improve the flow of wording - could be made more clearer.
Line 125 - add concentration of buffer and pH. assuming 1XPBS, pH7.4
Line 126 - covert rpm to "rcf" or "g"
Line 152 - Missing details on how to replicated this. Maybe add a reference for primers used or supplementary table, what Taq master mix and cycling conditions, size of amplicons. Did you sequence them to confirm if you used your own designed primers?
Line 153 - you have counted the expression but not actually sequenced the RNA. Make this clearer.
Line 174 - use italics for E. coli and add strain used.
Line 179 - Add "mouse" to indicate the species of the HEI-OC-1 cells.
Line 205 - add number of protein replicates.
Line 207 - add B-actin if purchased from same company and same concentration. if not, add details to Line 210. Out of interest, why switch reference gene from RNA to protein work?
Line 225. Missing a full stop
Line 368 - you mention p53 but do not have a statement on its expression from your nano string data - was it too unregulated?
Figure 2 - a subset of gene data is shown. I"m assuming that the other genes not shown did not show the same expression pattern as the nano string data. Otherwise, what was the basis of selection?
Line 404 - conclusion is too broad. You have clearly shown the 11 genes change significantly in their mRNA expression but only validated a subset and then focussed on IFNG. Thus, need to make this clearer.
Comments on the Quality of English Language
Minor corrections are required. See above suggestions.
Author Response
Author’s Reply to the Reviewer 1
Comments and Suggestions for Authors
I note that the figures the authors have presented are of high resolution but also provide the strong basis of their scientific argument which is clearly articulated in their discussion. I found the scientific argument to be clear and it was supported by appropriate references. Publishing this data would be beneficial to the field. The authors have identified the pitfalls in their final discussion paragraph. I agree with their conclusions (Lines 396-403).
Comments 1: [Line 29 - Use correct acronym – RT-qPCR. RT PCR refers to reverse transcription PCR]
Response 1: Thank you for your professional and helpful review. We revised the manuscript according to the reviewer’s comment (Line 29, 30, 122, 151, 152, 157, 241, 256, 259, 267, 268, 277)
Comments 2: [Line 30 - write microRNA in full or add to introduction - line 69.]
Response 2: Thank you for the valuable advice. We revised the manuscript according the reviewer’s suggestion (Line 30, 69)
Comments 3: [Line 32 - Remove a "s". e.g. Change "as age -related genes" to "age-related gene"]
Response 3: Thank you for your careful review. According the reviewer’s suggestion, the authors revised it (Line 32)
Comments 4: [Line 35 - remove "in vitro study" and add "in vitro to the end of the sentence]
Response 4: Thank you for your helpful review. Following the reviewer’s suggestion, the authors revised the sentence as follow: We found that the miR-409-3p down-regulates the expression of the IFNG in vitro (Line 35)
Comments 5: [Line 67 - I would not call it a GWAS as the cohort is too low and you have used RNA and not a genetic marker]
Response 5: Thank you for your careful review. According the reviewer’s suggestion, the authors revised the sentence as followed: This study performed analysis for RNA sequencing data from the ARHL patients and could find 11 genes related with ARHL (Line 67)
Comments 6: [Line 68 - add a reference or "this study"]
Response: Thank you for your careful review. According the reviewer’s suggestion, the authors revised the sentence as followed: This study performed analysis for RNA sequencing data from the ARHL patients and could find 11 genes related with ARHL (Line 67)
Comments 7: [Line 96 - remove space at the start of the sentence]
Response 7: Thank you for your careful review. According the reviewer’s suggestion, the authors revised the sentence (Line 96)
Comments 8: [Line 108 - use lower case "enrolled"]
Response 8: Thank you for your helpful review. Following the reviewer’s suggestion, the authors added “enrolled” to the sentence (Line 108)
Comments 9: [Line 113 - use "on the other hand" - i.e. remove the "s"]
Response 9: Thank you for your careful review. Following the reviewer’s suggestion, the authors revised the sentence (Line 113)
Comments 10: [Line 116 - alternative wording in the title? ie. mRNA differentiation expression assay, rather than just using mRNA assay]
Response 10: Thank you for your comment. As pointed out by the reviewer, the authors agreed using ‘mRNA differentiation expression assay’, rather than just using ‘mRNA assay’. So, the authors revised the sentence (Line 116)
Comments 11: [Lines 121-127 - improve the flow of wording - could be made more clearer]
Response 11: Thank you for your professional and careful review. As you pointed out, the authors revised the sentence as followed: Collected blood was gently mixed in EDTA tubes, and layered onto Ficoll-Paque (1:1 ratio). After centrifugation at 277 rcf for 30 minutes at room temperature without brake, the PBMC layer was collected, washed with 1X PBS (pH7.4), and centrifuged at 277 rcf for 10 minutes again. The supernatant was discarded, and the cells were resuspended in 1X PBS for RNA extraction (Line 122-126)
Comments 12: [Line 125 - add concentration of buffer and pH. assuming 1XPBS, pH7.4]
Response 12: Thank you for your careful review. According the reviewer’s suggestion, the authors added concentration of buffer and pH to the sentence as followed: After centrifugation at 277 rcf for 30 minutes at room temperature without brake, the PBMC layer was collected, washed with 1X PBS (pH7.4), and centrifuged at 277 rcf for 10 minutes again. The supernatant was discarded, and the cells were resuspended in 1X PBS for RNA extraction. (Line 123-126)
Comments 13: [Line 152 - Missing details on how to replicated this. Maybe add a reference for primers used or supplementary table, what Taq master mix and cycling conditions, size of amplicons. Did you sequence them to confirm if you used your own designed primers?]
Response 13: Thank you for your careful review. According the reviewer’s suggestion, Information on reaction components is attached. We created a mix with a total volume of 10 µL. So, the authors added sentence as followed: To perform RT-qPCR, mix 2 µL of 5x miscript HiFlex Buffer, 1 µL of 10x miScript Nucleics Mix, 1 µL of miScript Reverse Transcriptase Mix, and 1 µg of Template RNA. Then, mix RNase-free water to make a total volume of 10 µL. (Line 155-157)
Comments 14: [Line 153 - you have counted the expression but not actually sequenced the RNA. Make this clearer]
Response 14: As you pointed out, we revised “Validation of target gene expression” as followed: To validate the expression of candidate genes identified from mRNA sequencing results, RT-qPCR was employed with whole blood of the NH and ARHL patients (Line 152-153). Additionally, we revised “Results” as followed: Finally, among 11 genes, five genes showing correlation with RNA sequence were found (Line 241-243)
Comments 15: [Line 174 - use italics for E. coli and add strain used]
Response 15: Thank you for your careful review. According the reviewer’s suggestion, the authors revised the sentence. We modified E. coli in italics and added strain DH5α (Line 176)
Comments 16: [Line 179 - Add "mouse" to indicate the species of the HEI-OC-1 cells]
Response 16: Thank you for your helpful review. Following the reviewer’s suggestion, the authors revised the sentence as followed: The House Ear Institute-Organ of Corti 1 (HEI-OC-1) cells of mouse were incubated at 37°C with a 5% concentration of carbon dioxide and harvested 10% FBS DMEM (Dulbecco’s Modified Eagle’s Medium, Welgene, Republic of Korea) (Line 181)
Comments 17: [Line 205 - add number of protein replicates]
Response 17: Thank you for your careful review. Following the reviewer’s suggestion, the authors added number of protein replicates: The following protein replication was repeated three times. (Line 207)
Comments 18: [Line 207 - add B-actin if purchased from same company and same concentration. if not, add details to Line 210. Out of interest, why switch reference gene from RNA to protein work?]
Response 18: Thank you for your professional and helpful review. According the reviewer’s suggestion, the authors revised the B-actin’s information (Line 212). And, thank you for your professional opinion and interest in our topic. Initially, we validated the target gene with RT-qPCR of whole blood sample and confirmed consistent results with Nanostring analysis. Finally, western blot was used to confirm similar results of RT-qPCR at the protein level.
Comments 19: [Line 225. Missing a full stop]
Response 19: Thank you for your careful review. According the reviewer’s suggestion, the authors added a full stop as followed: Figure 1C portrays a volcano plot representing the expression levels of candidate genes, while Figure 1D presents a scatter plot. (Line 228)
Comments 20: Line 368 - you mention p53 but do not have a statement on its expression from your nano string data - was it too unregulated?
Response 20: Thank you for the valuable advice. We agreed your professional opinion and deleted these statements and reference of p53.
Comments 21: [Figure 2 - a subset of gene data is shown. I"m assuming that the other genes not shown did not show the same expression pattern as the nano string data. Otherwise, what was the basis of selection?]
Response 21: Thank you for your professional and careful review. The basis of selection is the correlation between RNA sequencing results and the RT-qPCR value of whole blood samples (normal hearing and aging related hearing loss patients). So, we revised the manuscript as followed: To validate the mRNA assay results of selected DEGs, RT-qPCR was conducted using whole blood samples. Finally, among 11 genes, five genes showing correlation with RNA sequencing results were found (Line241-243)
Comments 22: [Line 404 - conclusion is too broad. You have clearly shown the 11 genes change significantly in their mRNA expression but only validated a subset and then focussed on IFNG. Thus, need to make this clearer]
Response 22: As you pointed out, the authors revised the conclusion as followed: We suggest that the potential role of miR-409-3p regulates IFNG expression by the p16 signaling pathway in ARHL patients. These results indicated that distinct developmental strategies for preventative drugs or methods are needed to tackle hearing loss by aging (Line408-410)

Reviewer 2 Report
Comments and Suggestions for Authors
miR-409-3p regulates IFNG and p16 signaling in human blood of aging
related hearing loss
Topic is very interesting. Study evaluates to evaluate the potential as a protective tool for age related hearing loss (ARHL) treatment by comparing human blood-based target gene-miRNA associations regulated in ARHL. The roles of hsa-microRNA (miR)-409-3p in senescencewas already explored (Lee YN, Wu YJ, Lee HI, Wang HH, Hung CL, Chang CY, Chou YH, Tien TY, Lee CW, Lin CF, Su CH, Yeh HI. Hsa-miR-409-3p regulates endothelial progenitor senescence via PP2A-P38 and is a potential ageing marker in humans. J Cell Mol Med. 2023 Mar;27(5):687-700). This case control study included a limited number of patients but methods used for target validation and to quantify gene expression are pertinent. It also elucidate a signalling patway. Cell culture and transfection, western blot assay and statistical analysis were used to obtain a mRNA-assay heatmap and to predict the 3′UTR sequences of IFNG gene with their binding sites for miR-409-3p. All results are presented in five figures. Conclusion is in accordance with the obtained results. References must be up-dated.
Author Response
Author’s Reply to Reviewer 2
Comments and Suggestions for Authors
Topic is very interesting. Study evaluates to evaluate the potential as a protective tool for age related hearing loss (ARHL) treatment by comparing human blood-based target gene-miRNA associations regulated in ARHL. The roles of hsa-microRNA (miR)-409-3p in senescencewas already explored (Lee YN, Wu YJ, Lee HI, Wang HH, Hung CL, Chang CY, Chou YH, Tien TY, Lee CW, Lin CF, Su CH, Yeh HI. Hsa-miR-409-3p regulates endothelial progenitor senescence via PP2A-P38 and is a potential ageing marker in humans. J Cell Mol Med. 2023 Mar;27(5):687-700). This case control study included a limited number of patients but methods used for target validation and to quantify gene expression are pertinent. It also elucidate a signalling patway. Cell culture and transfection, western blot assay and statistical analysis were used to obtain a mRNA-assay heatmap and to predict the 3′UTR sequences of IFNG gene with their binding sites for miR-409-3p. All results are presented in five figures. Conclusion is in accordance with the obtained results.
Comments 1: [References must be up-dated]
Response 1: Thank you for your thorough review of our manuscript. We appreciate your interest of our topic. The reference you mentioned is already written in Line 546, reference 34.